# Crop Classification of Satellite Imagery Using Synthetic Multitemporal and Multispectral Images in Convolutional Neural Networks

**Guillermo Siesto** , **Marcos Fernández-Sellers** and **Adolfo Lozano-Tello** *

Quercus Software Engineering Group, Universidad de Extremadura, 10003 Cáceres, Spain; guillermosiesto@unex.es (G.S.); marcosfs@unex.es (M.F.-S.)
* Correspondence: alozano@unex.es

**Abstract:** The demand for new tools for mass remote sensing of crops, combined with the open and free availability of satellite imagery, has prompted the development of new methods for crop classification. Because this classification is frequently required to be completed within a specific time frame, performance is also essential. In this work, we propose a new method that creates synthetic images by extracting satellite data at the pixel level, processing all available bands, as well as their data distributed over time considering images from multiple dates. With this approach, data from images of Sentinel-2 are used by a deep convolutional network system, which will extract the necessary information to discern between different types of crops over a year after being trained with data from previous years. Following the proposed methodology, it is possible to classify crops and distinguish between several crop classes while also being computationally low-cost. A software system that implements this method has been used in an area of Extremadura (Spain) as a complementary monitoring tool for the subsidies supported by the Common Agricultural Policy of the European Union.

**Keywords:** remote sensing; crop classification; convolutional neural networks; sentinel-2; satellite imagery; multi-spectral; multi-temporal; common agricultural policy



## 1. Introduction

Throughout history, there has always been a strong interest in the status of the land. With suitable information, it is possible to calculate an area's potential wealth based on the products that could be produced from its land, both in terms of extent and its quality. These processes, until recently, have had to be carried out by manually collecting samples and studying the changes over time. Over the years, there has been an increase in the development of technologies to help address this issue, and recent projects such as the Sentinel-2 missions have come to the fore assisting in the development of the classification algorithms developed to date. Thanks to these satellite imagery technologies, updated multispectral samples are obtained automatically.

Due to the current decay of the rural world, there is an attempt to encourage farmers to work the land by providing subsidies to keep the land tended and sown. In 2014, the Directorate-General for Agriculture and Rural Development of the European Commission introduced new regulations [1] in the field of automatic land control and especially in the field of agriculture. They proposed the procedure of regular and systematic observation, tracking and evaluation of all eligibility criteria, commitments and other obligations which can be monitored by Copernicus Sentinel satellite data or equivalent, over a period of time that allows conclusions to be drawn on the eligibility of the aid or support requested where necessary. Therefore, the European Union (EU) Common Agricultural Policy (CAP) paying agency of the Regional Government of Extremadura, Spain, is starting to take an interest in the development of automatic tools for the control of crops in the region, such as detecting

fraud for agricultural subsidies or providing farmers with tools to keep track of their land and improve their production. In keeping with this goal, the objective of this work has been to develop a new approach for identifying crops using synthetic images derived from satellite data. The method was applied to control the CAP with summer crops planted in an area of Extremadura, Spain. The software system that implements the method trains with data from previous years and predicts with data from the following year.

The development of classification methods for these problems has mainly involved machine learning using random forest [2–4] but over time, other methods like neural networks have been contrasted [2,3], and even recurrent neural networks have been tested [5] with good results. Studies such as [6–8] analyse the different types of related articles to date concluding that deep learning, in particular convolutional neural networks (CNN), offers better performance than other methods.

Neural networks applied to satellite images are one of the most commonly used techniques for land classification. Most of these projects look at a single sentinel image with its multispectral information and train from it with segmentation [9] even to classify crops [10], making the prediction for the same year. Even pre-trained models have been tested to make these segmentations, concluding that the best methods in terms of time and accuracy are deep convolutional networks [11]. Many papers have been published in the field of crop classification research [12,13]. These methods have mainly involved multispectral information, but other efforts [14–16] have shown the benefits of adding a multitemporal variable to these first data for crop classification, as you can access the data progressively at different times of the year. In some other cases, these approaches use only the time variable with indicators such as Normalised Difference Vegetation Index (NDVI) or Enhanced Vegetation Index (EVI) [17–21] with substantial results.

Following a study of the state of the art, the CAP paying agency of the Regional Government of Extremadura has decided to develop classification models using temporality for its crop identification procedures to control the aid it gives to farmers in the region. Seasonality must be considered because it is needed to identify crops of the present year knowing only what has been sown in previous years and, therefore, it is also important to have data on more than one year, due to weather complications. As the most promising results for the classification of satellite images currently involves deep neural networks, the development of a convolutional neural network has been considered with the intention of prioritising accuracy and low computational effort thanks to a new input methodology, and to have these systems in place before the payment campaign that takes place every year at the end of the summer, when the aid to be given will be considered.

This paper reflects the progress made in this approach and will be divided as follows: Section 2 presents the proposed method based on multi-temporal and multi-spectral imaging as well as the collection and processing of data; Section 3 describes the chosen neural network architectures adopted, how they have been applied in the Extremadura region (Spain) and the results of the 2020 campaign. Finally, Section 4 shows the conclusions and possible applications of the proposal.

## 2. Method

Our proposal is a method based on multi-spectral and multi-temporal synthetic inputs to classify crops from satellite images through CNNs. It uses synthetically generated images as inputs with all the bands of the Sentinel-2 spectrum and information from several months. In other words, these synthetic inputs had two dimensions: in one dimension there were the 12 spectral bands of Sentinel-2, and in the other, the temporal variables of each image that had been collected over time. As described in the previous section, other methods of crop classification have, to date, trained with one or very few images, looking at the whole image pixel by pixel, and analysing the regions of the images as the convolutions are made. In this proposal, we extracted the information from each of the relevant pixels of the crops and their evolution over time, to create synthetic images for each of the pixels of the crops. That mix was the input for the convolutional network.

Due to the crop monitoring approach, it was important to monitor as many dates as possible, because of (partial or total) crop abandonment or anomalous situations that arise. Furthermore, it also helped to reduce noise due to atmospheric conditions, such as clouds, that might occur in some areas on certain dates. Some date periods were more significant for the identification of specific crops than others, so it was important to use a substantial number of images from different dates.

As we referenced in the previous section, other approaches have tried to add another temporal dimension directly on the Sentinel images using only a few days, which increases the complexity of an already big input. As our aim was to speed up the training processes, working with these inputs would be very costly if the number of images was massive, as a large part of the pixels in the images will not be worked on due to the lack of label information on some parts of them. For these reasons, a new synthetic matrix was created, to obtain similar information, but in a much simpler input so that the networks could operate more quickly and easily. It should also be noted that all the pre-processing of each pixel would be done only once before the training, while the training could be done many times until a good solution was found. In the following sections, we describe the steps for applying our proposed method.

### 2.1. Data Collection and Processing

The first step was the generation of the inputs for the learning of the CNN. As mentioned, we used the 13 spectral bands of all Sentinel-2 images for a period within a year, grouping the information at the pixel level. Although we proposed to use the largest number of images (throughout the year) with all the information from the entire satellite bands, the generation of this synthetic input could be adapted according to the needs of each case.

Data collection was obtained using the Sentinel-2 satellites thanks to the European Space Agency (ESA). These data contained 13 data points for a day, consisting of each one of the spectral bands. One of these bands, the B10, was lost after atmospheric correction of the image [22], so in the end, 12 bands remained to work with. The process of downloading and processing the images had previously been investigated in another work in which the geometric situation was linked to the Sentinel-2 data [23].

#### 2.1.1. Processing Geometry and Plots

In order to identify the pixels of interest, the first process was to start looking for matches between the geographical data of the plots and the satellite data to obtain the information from each of the pixels that included the plots.

After the geolocation of each of the edges of the plots was available, specific algorithms were used to extract the pixels enclosed by their area. As can be seen in Figure 1, we worked with a mesh that contained the geolocation information. This process was carried out for each of the images in the time period used to generate the synthetic input, with the information on each of the plots used later to make the inputs of the network. Due to the resolution of Sentinel-2 images, the pixels represented had a size of 10 × 10 m, each with its own multispectral information. It should be noted that the resolution of some bands ranged from 10 m, 20 m to 60 m. As a result, tools were used to resample lower resolution bands to higher resolution bands, giving all of them a resolution of 10 × 10 m.

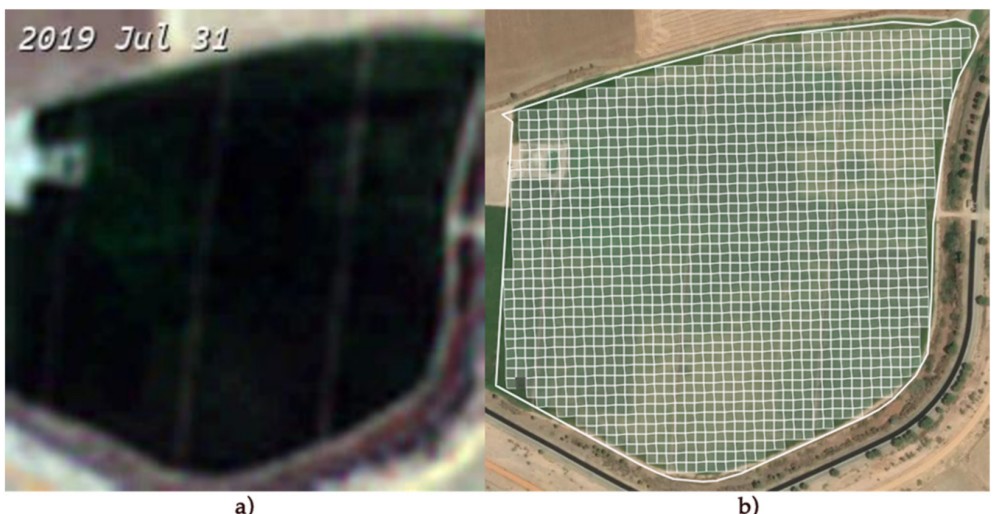

**Figure 1.** Pixel grid of a corn crop. Image (**a**) is an image extracted from Sentinel-2 as of 21 July 2019; Image (**b**) represents the pixels of a plot.

### 2.1.2. Atmospheric Conditions and Potential Noise

The images obtained by the satellite were not always reliable. Meteorological factors can greatly affect the data obtained, adding erroneous information that can contaminate the results. This is why a cloud detection system previously developed by ESA, which makes use of neural networks and categorises conditions into Snow, Fine Cirrus (Type of cloud), High probability of cloud, Medium probability of cloud, Unclassified, Water, No vegetation, Vegetation, Surplus due to cloud, Very dark area, Saturated or defective, and No data was used based on the algorithm proposed in [24–26] Figure 2.

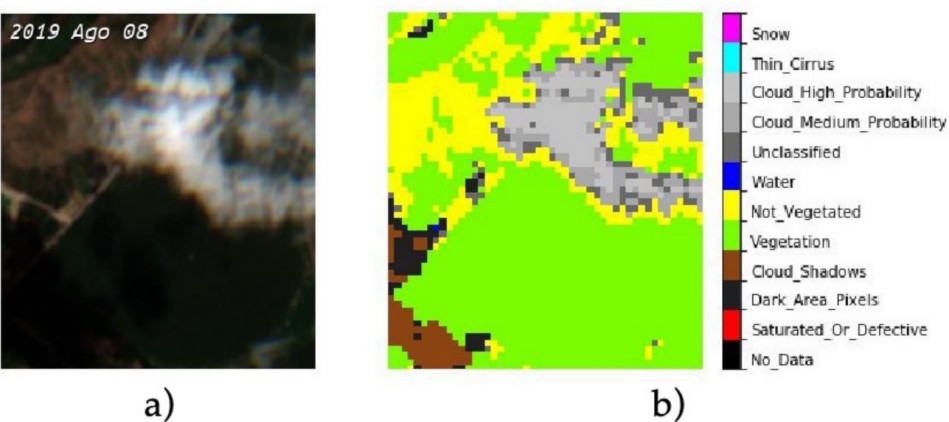

**Figure 2.** (**a**) Sentinel 2 RGB image sample with a (**b**) mask after the output of the atmospheric variable detection model provided by the European Space Agency (ESA).

All data was obtained directly from the API provided by the Sentinel satellites. For the treatment of these data, when it came to finding anomalies, ESA's own models were used for the detection of clouds, water, shadows, etc. In addition, in order to obtain better results, our own outlier detection algorithms were used and interpolation was performed on all the data. This was done as the anomaly detection models discussed earlier are not 100% reliable. As all data were treated equally, the noise was reduced with a negligible loss of information. Even after this process was carried out, outliers may remain due to the accuracy of the models provided by ESA, for example, cloud shadows were quite difficult to detect. For this reason, an adaptation of the algorithm called Hampel Filter [27], which removes any outliers and smooths the data, was run at the end.

### 2.1.3. Time Period Selection

As mentioned, the inputs to the CNN were the synthetic images based on the information from the pixels from all bands of the Sentinel images during several months in one year. As can be seen in an earlier experiment [23], the more years the network is trained with, the more accurate it is expected to be.

The Sentinel satellite orbits cause new images to be obtained every 5 days (every 3 days in the case of overlapping UTM time zones) [28]. As can be seen in Figure 3, there were differences between different years, which is why it was interesting to study more than one year to avoid the possibility of the network being biased due to temporality. In addition, it was important to emphasise that the chosen period represented by the dashed lines (From 1 April to 15 July) was chosen on the grounds of expert criteria, as earlier data was poor and not be meaningful due to the rainy season, and with the knowledge that aid for a land control system would be given at the end of the summer, so classification had to be done in advance. In other types of research, seasonality is not an important variable, as the classification of urban area, land, water, etc. is not as time-varying as a crop harvest.

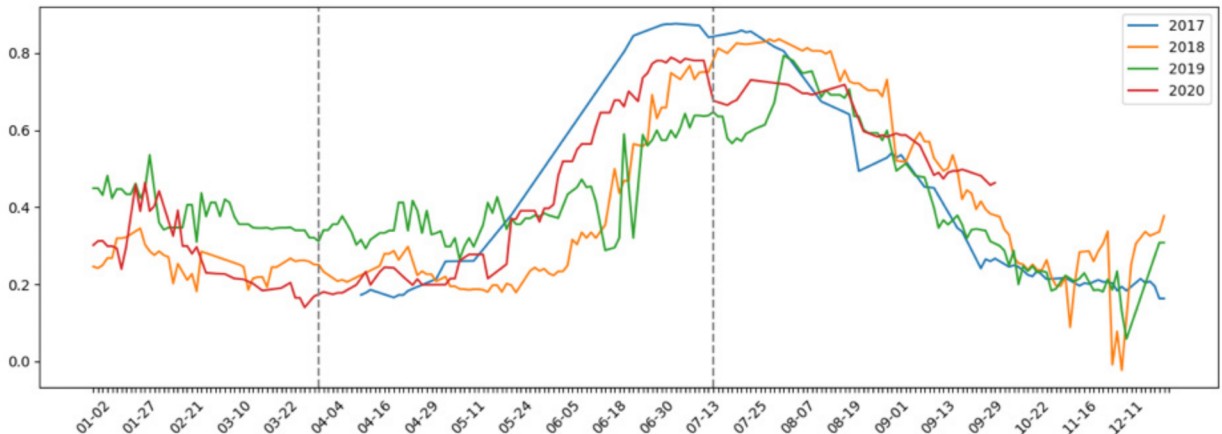

**Figure 3.** Representation of corn Normalised Difference Vegetation Index (NDVI) over 4 years. The two dashed lines represent the study period chosen to generate the network input.

The minimum update period for satellite data is 5 days, although some areas where there is an overlap of two orbits have an extra day between those 5 days. Nevertheless, the minimum period considered was 5 days, with an average if more data was available. All elements were interpolated due to missing days and atmospheric conditions, in order to reduce any noise or outliers in the data. On the other hand, having leap years, where the days in certain years do not match exactly, was another reason why the data is treated 5 days at a time (knowing that this is the maximum possible time between new data). These small periods are referred to as pentads.

Another consideration to take into account is that in 2017, there was only one satellite in orbit, so there was an update every 10 days. In order to work in the same way as for other years, information was interpolated every 5 days, slightly sacrificing the precision of the data at the cost of being able to use information from that year.

### 2.1.4. Creating Synthetic Images

With the complexity of the data due to taking into account the temporality during a year, it was decided to create a new type of input instead of taking one single Sentinel-2 image of a single year and analysing it, as has been done so far in most investigations [9,10]. The information was therefore gathered from different days of the year during the corresponding period. With this information at the pixel level, a matrix combining spectral data with temporal data was created and used as the input to the neural network. A similar approach was used for the classification of time series datasets using the relative

position matrix [6], which reflected the benefit of using two-dimensional data structures to reduce the complexity of the original data, making the networks better able to differentiate between classes when training.

Each of the pixels of the raw images was analysed one by one with the pixel information extracted from them by adding the variable of temporality. In this way, a matrix was created for every pixel, with information on each of the crops covering the whole period.

In Figure 4, on the left of the diagram is the RGB image of one area of 5000 × 5000 m. Looking at a single pixel depicted in green, this image was taken over a period of time, specifically 21 occurrences, from the 1st of April to the 15th of July. As each of the images contained information from the 12 frequency bands (B1, B2, B3, B4, B5, B6, B7, B8, B8A, B9, B11, B12), this data was extracted for each of the dates to generate the input to the neural network (represented on the right of the image). Values of the entries shown in light green are close to one and those shown in dark blue are close to zero.

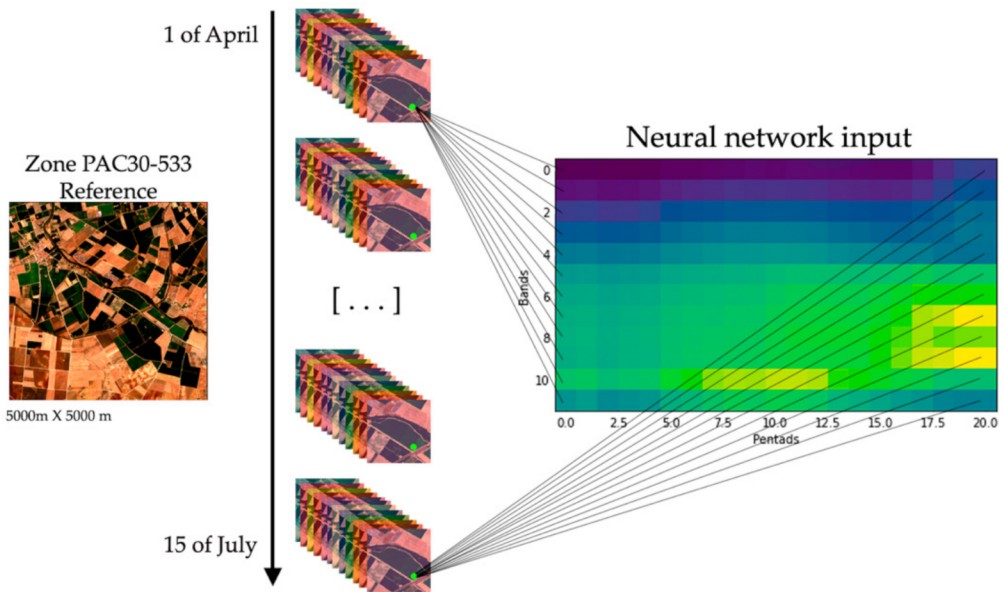

**Figure 4.** Representation of the extraction of the band and temporal information for each of the neural network inputs.

With both temporal data and different frequency bands, the network input could be organised as a matrix.

- Dimension 1: Time series of pentads (5-day periods), from 1 April to 15 July, 21 pentads. With each of these pentads, there were 21 data points.
- Dimension 2: Different types of frequency bands of the Sentinel 2 satellite. One dimension represents the satellite bands and the other their variation over time.

### 2.2. Training and Testing of CNN

The models used for this project consisted of convolutional neural networks [29]. CNNs are a type of neural network designed primarily to deal with image data as they perform best on classification problems. The model learns to automatically extract the most important features from the data in order to discern them among others. They consist of multiple layers of convolutional filters of one or more dimensions, to finally derive in a fully connected network. These two stages are known as the learning phase and the classification phase.

The learning phase uses reduction or sampling, where the input matrix is converted into different small or large shapes in each dimension, which has its own characteristics extracted from the previous layer. The classification phase, corresponding to the completely connected parts, focuses on classification, concentrating its efforts on providing one of the

outputs between the final classes. Each of the network nodes in each of its layers has its own activation function [30], which influences the output of each of the layers depending on the type of activation used. Moreover, between the layers, systems such as drop-out [31] or batch-normalization [32] are applied to facilitate learning and to avoid factors such as overtraining, which would be the memorisation by the network of the inputs instead of learning about their most differentiating features.

All these different configurations were continuously modified and tested according to the type of problem being dealt with. In the following section, the elements chosen for the generalisation of the convolutional network can be seen in more detail.

## 3. Application of the Method to the Region of Extremadura, Spain

The method described in the previous section was used as a support tool by the Regional Government of Extremadura in the crops identification programme declared in the European Union Common Agricultural Policy (CAP). It was particularly suitable because a large amount of plot information was available for several years to feed the CNN. This section describes how the method was applied to identify seven crop groups in 2020 with CNN models trained on 2017, 2018 and 2019 synthetic inputs. The area where it was applied, the type of data obtained, data treatment, training with the convolutional networks, and the analysis of the results are explained in the following sections.

### 3.1. Work Area

The proposed method used a large amount of information to process the data with the evolution of each pixel, so it was important to optimise the memory. The original Sentinel images were very large, which would lead to information processing problems due to RAM memory as well as storage, as many pixels would be saved from outside the interested area. That is why it was decided to divide the earth map using the same coordinate system as the Sentinel satellites (SRC) EPSG:32630. WGS 84/UTM zone 30N y 29N. A division size of 5000 × 5000 m was chosen to dissect the region. As is visible in Figure 5, there were two types of inclinations in the zones. This was due to the different zone coordinates system (29N and 30N) [28].

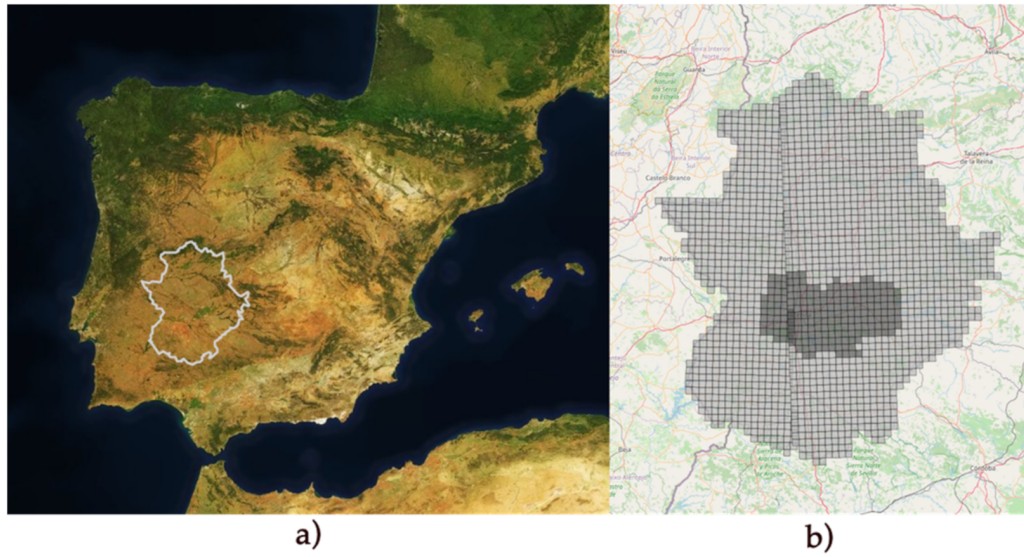

**Figure 5.** (**a**) The Iberian Peninsula; (**b**) The light grey area is the entire region of Extremadura, and the dark grey area is the work area (region of Mérida and Don Benito, Spain).

The satellite images were obtained through the API offered by the ESA, transforming the 110 × 110 km size to slices of 5 × 5 km (5000 × 5000 m) as represented in Figure 5b. As each of the zones covered 25 square kilometres and there were 204 zones, the total

number of hectares to be processed was 510,000 ha. For all of them, a prediction was made regardless of whether it was a ground truth declared area or not, to check predictions of possible adjoining plots.

### 3.2. Data Collection

Information on the situation of the parcels in the areas chosen was needed for the collection of a dataset that had the labels of each of the crops for each of the pixels in the plots. This information was provided by the Regional Government of Extremadura which records the entire situation of the plots of land in the region for CAP control, thanks to the declarations made by farmers to official bodies and to the random checks undertaken every year. These data, together with the geographical data, were treated with algorithms linked with the satellite data, which led to a set of labelled information. An image with the geographical representation of all the plots studied can be seen in Figure 6. The hectares declared within this area totalled 34,135 ha.

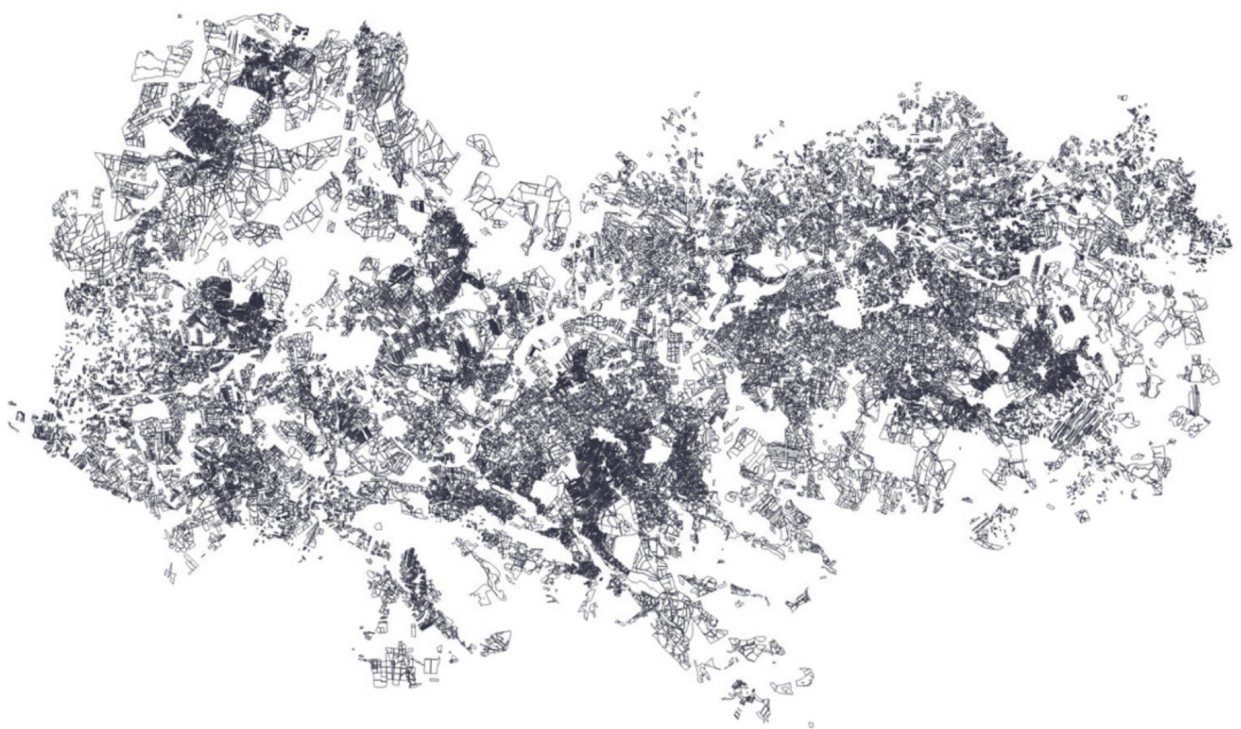

**Figure 6.** Geographical representation of the geolocation of the study plots. Region of Mérida (38°54′ N 6°20′ W) and Don Benito (38°57′16.3″ N 5°51′42.2″ W), Spain.

### 3.3. Crop Survey and Categories

The paying agency of the Regional Government of Extremadura collects the polygons and their corresponding crops for the CAP. They group the crops into the following categories: corn (corn, sweet corn), tomato (tomato, tomato for processing), rice, sunflower, potato, onion/garlic/leek, asparagus.

This choice of crops has been based on the criteria of the amount of data for the area given to this agency and the subsidies it gives to the agricultural industry. Although there are more crops in this area, they are not sufficiently representative. The seven crop groups chosen, correspond to those grown in the Mérida-Don Benito area during the summer season.

As discussed in Section 2.1.4, the inputs of the neural network were synthetic images created with the 12 bands of the Sentinel-2 spectrum and their evolution over time per pixel. For each crop, the neural network identified the pattern of the evolution in time of the values of these bands. Each of the crops had its own differentiating elements that made

the network classify according to each of the outputs. Figure 7 shows an average of all the values for each of the crops for the year 2019. The greenest ones represent values of the inputs closest to one, and the bluest ones are closest to zero. In this way, it can be seen at a glance that the average of each of the results in the same year shows substantial differences for each of the crops.

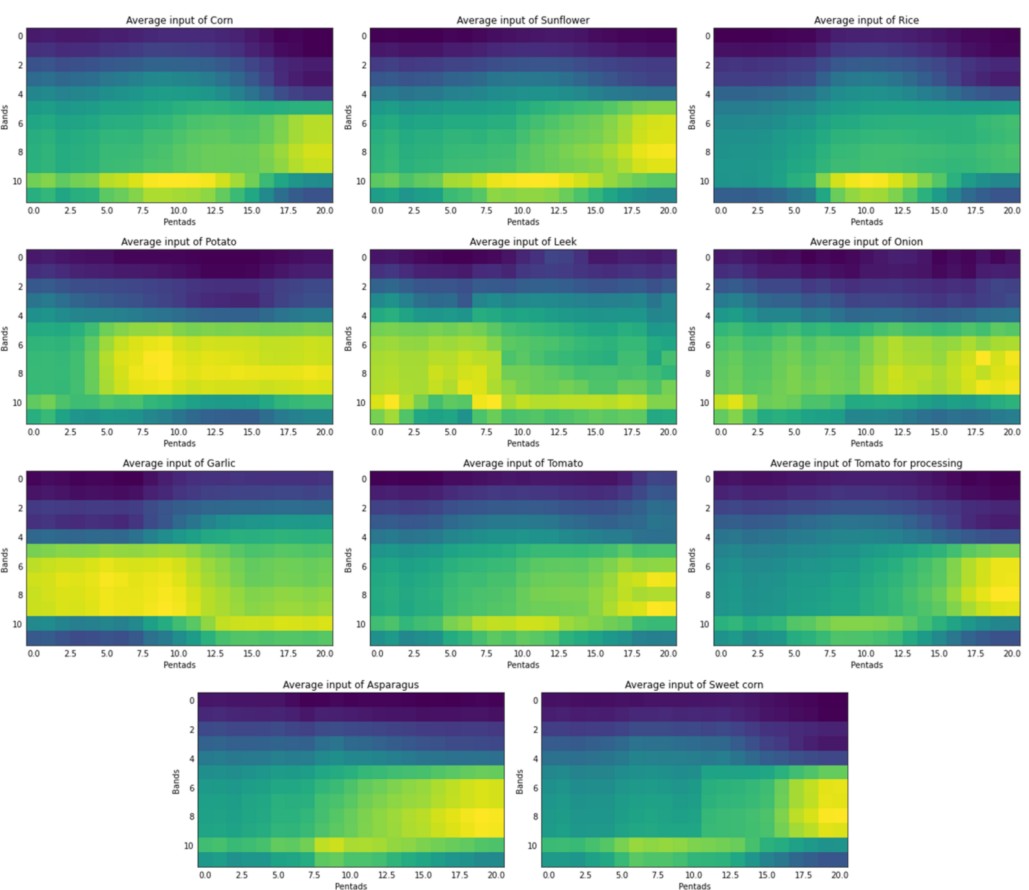

**Figure 7.** Average input of each of the studied crops. Representation of the 2019 average of training entries per crop.

### 3.4. Training and Testing Sets

The used datasets consisted of information from the seven crop groups throughout the years 2017, 2018, 2019 and 2020, with the first three years being used for training/validation and 2020 used for predicting and testing. This was done because it was desired to design a crop control system that could be used to predict information from a year without having knowledge of the labelled state of the land at that time, only having the labelled data from previous years, for monitoring crops to manage the CAP aids.

As can be seen in Figure 8, the data was imbalanced, so a weight generation function was also used to compute the balanced weight for each of the classes for the training.

The distribution between training, validation and testing is shown in Table 1. As the results were tested for 2020 without using this year as training, the whole of 2020 was used as a test. The training was 85% and the validation 15% equally for each of the three previous years.

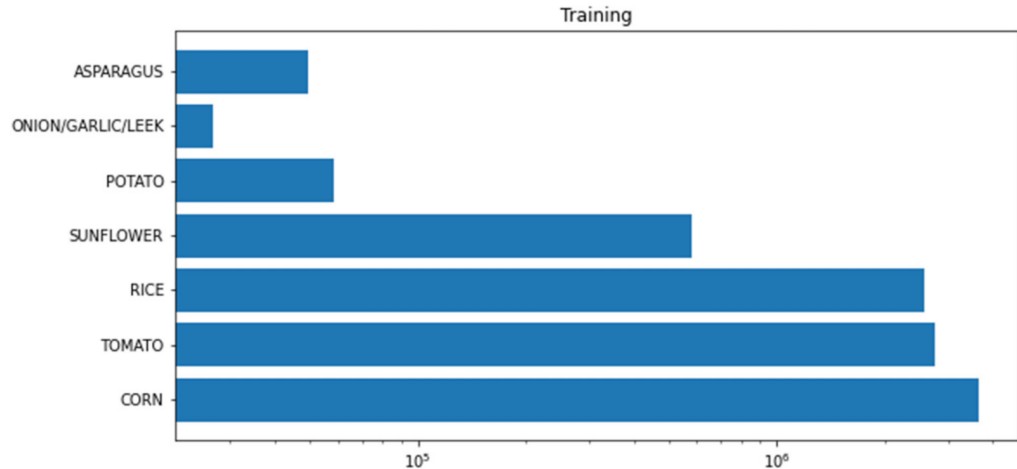

**Figure 8.** Distribution of training data considering the years 2017, 2018 and 2019. The axis of x-coordinates in base-10.

**Table 1.** Distribution of datasets in training, validation and testing.

| Year | Train | Validation | Test | Total per Year |
|---|---|---|---|---|
| 2017 | 3,763,045 | 664,060 | - | 4,427,105 |
| 2018 | 3,719,405 | 656,360 | - | 4,375,765 |
| 2019 | 2,225,530 | 392,733 | - | 2,618,263 |
| 2020 | - | - | 2,382,494 | 2,382,494 |
| Train/Validation | 9,707,980 (85%) | 1,713,153 (15%) | - | 11,421,133 |
| Test | - | - | 2,382,494 (100%) | 2,382,494 |

*3.5. CNN Design*

Once the datasets were obtained and processed, the convolutional networks were trained and tested. First, a network design already implemented in similar problems was used as a starting point, training this architecture, and compiling the results. We then went on to design a new architecture more adapted to our specific system in order to try to obtain better results and, above all, to gain in execution time, since the advantage of synthetic inputs meant that the models did not have to be so complex.

One of the first trials was based on a typical model explained in a paper [3] called the Convolutional Neural Network variant. It must be taken into account that this model was not designed for the same methodology, and it naturally does not obtain the best results in terms of accuracy and, above all, time. It consisted of four convolutional layers with a hyperbolic tangent activation function (TanH). Batch-normalization was applied in two-by-two layers, ending in two dense layers. Dropout was also used between the layers of the net. The last dense layer ended in a Softmax activation with the expected number of outputs.

On the other hand, the design proposed for this problem may vary to some extent. From the $21 \times 12$ input onwards, two convolution phases were applied, one by increasing the filters and the other by decreasing them. Between each of the convolutions, Batch-normalization and dropout were applied to avoid overfitting. Finally, a flatten was linked to a dense layer with the stipulated number of outputs. The activation of the convolutional layers was a rectified linear unit activation function (ReLU) and on the dense layer Softmax. During the whole process, the early-stopping technique was used, which monitored the progress of the loss values to avoid overtraining the network by stopping the training at an appropriate time. The activation function layers, filtered sizes as well as the dropout, were chosen on a trial-and-error basis by running the nets with different values and using expert judgement several times. The Keras library in Python 3 was used for this project, which

made use of TensorFlow at a low level. The configuration and structure of the network are visually represented in Figure 9.

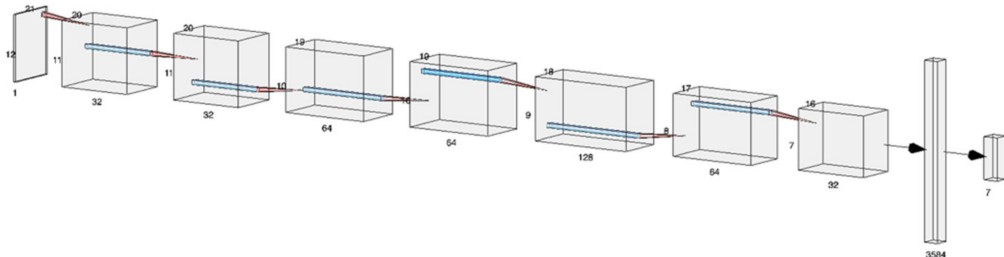

**Figure 9.** Schematic representation of the different layers of the convolutional network model developed for this project.

### 3.6. Results and Discussion

Our models were used to identify the Extremadura 2020 CAP with the proposed model based on our neural network, but we also carried out the training process with a classic CNN to be able to compare the results. Following the training sessions came the test phase where the data corresponding to the information obtained in 2020 was passed on. After training the two models with the same data, we obtained the results with the prediction of the test (2020) for the 204 zones, represented in Table 2.

**Table 2.** Average benchmarking between the different executions.

| Model | Accuracy | F-Score [33] | Epochs | Training Time [1] |
|---|---|---|---|---|
| Classic CNN | 95,116 | 0.95 | 22 | 38 h 37 min (6280 s avg/epoch) |
| Proposed model | 96,229 | 0.96 | 10 | 3 h 39 min 30 s (1317 s avg/epoch) |

[1] The training has been processed using a graphic card GeForce GTX 1060 3GB, 450.102.04, 3011 MiB.

After passing the test data to the models, the confusion matrices and their corresponding reports in terms of Accuracy, Loss and F-Score were obtained as shown in Figures 10 and 11. It can be seen that in both models, the crop groups with the best results were the largest (as can be seen in Figure 8) with the incorrect results generally concentrated and assigned to the Corn, Tomato and Sunflower groups.

To provide a more detailed overview of the obtained accuracies and losses, Table 3 represents 12 of the 204 zones, sorted in descending order by the number of pixels.

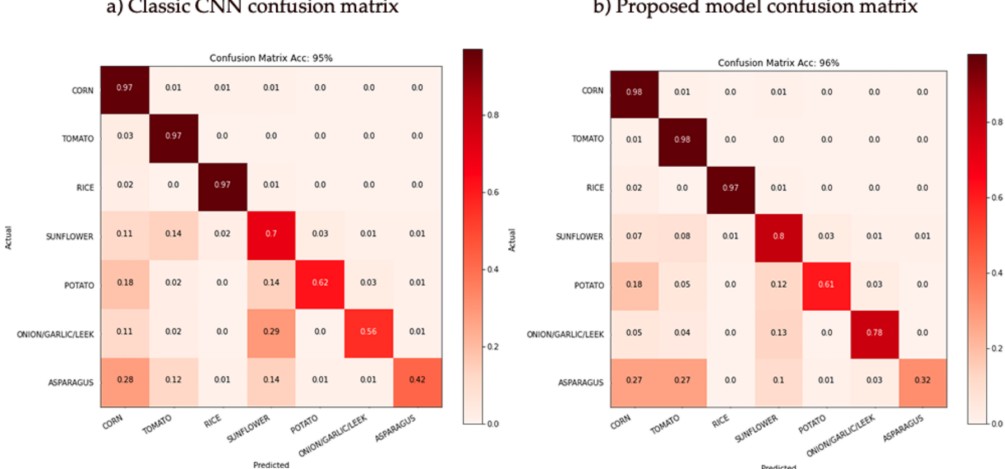

**Figure 10.** Comparative confusion matrix of the two model architectures. (**a**) Classic convolutional neural network (CNN) confusion matrix; (**b**) Proposed model confusion matrix.

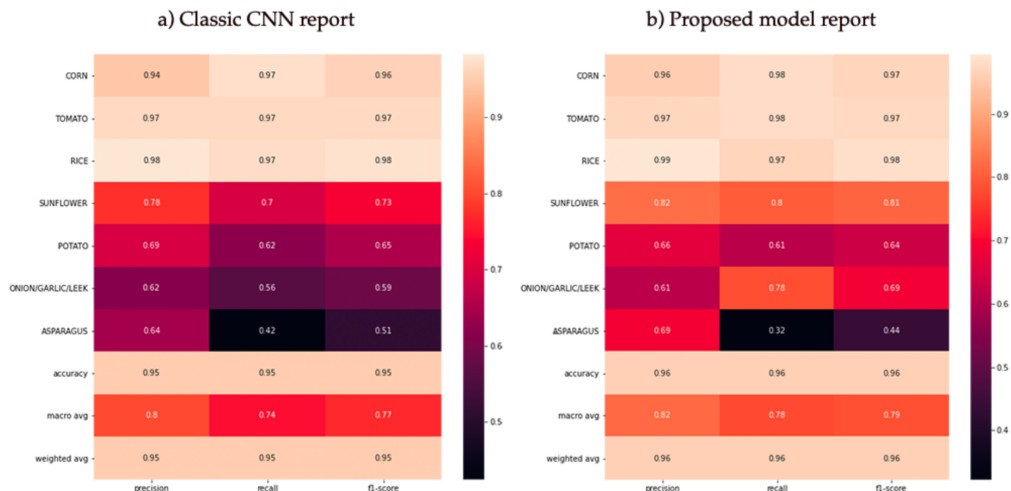

**Figure 11.** Comparative report of the two model architectures. (**a**) Classic CNN report; (**b**) Proposed model report.

**Table 3.** Results of the first 12 zones ordered by the highest number of pixels in the studied area with the proposed model predictions.

| Pixels Analysed | ID Zone | Accuracy | Loss |
|---|---|---|---|
| 100,318 | Zone 30-581 | 0.9737 | 0.1188 |
| 96,556 | Zone 30-532 | 0.9622 | 0.1525 |
| 90,651 | Zone 30-680 | 0.9714 | 0.0856 |
| 83,454 | Zone 30-531 | 0.9560 | 0.2377 |
| 81,895 | Zone 30-480 | 0.9748 | 0.1236 |
| 81,015 | Zone 30-369 | 0.9602 | 0.2404 |
| 79,051 | Zone 30-368 | 0.9657 | 0.1064 |
| 78,974 | Zone 30-530 | 0.9619 | 0.1507 |
| 77,202 | Zone 30-424 | 0.9717 | 0.0817 |
| 74,292 | Zone 30-425 | 0.9724 | 0.0942 |
| 74,169 | Zone 30-370 | 0.9818 | 0.0619 |
| 70,968 | Zone 30-679 | 0.9731 | 0.1125 |
| ... | ... | ... | ... |

As can be seen, the two networks obtained remarkable average results, with the biggest difference in training time due to the adaptation of the proposed network to the new types of synthetic image inputs explained in Section 2.1.4, which considerably reduced the size of the network. It should be emphasised that these results cannot be compared with the ones reported in the referenced paper, as they do not take the time variable into account and are not developed on the same types of data, both in the terrain and in the categories. Only the network architecture has been used as a reference for the state-of-the-art. In Table 3, which shows the areas with the most pixels broken down, it is visible that the accuracy results are very high, as they are covered by most of the crops. However, when looking at the crop-by-crop accuracies, it can be seen that the three minority crops do not offer comparable accuracy, with asparagus being the one that offers the worst results, as can be seen in the confusion matrix in Figure 10. From an agricultural perspective, these three types of crops, apart from being in the minority, are also less bushy than corn, rice, tomatoes, or sunflowers, as they are bulbs or grow at ground level, and do not have large leaves. It should be remembered that the precision of the satellites used is 10 by 10 metres, so this could be quite a substantial factor.

### 3.7. Visual Analysis of the Results

A mask view of the studied plots in the year 2020 is provided for a visual check of the results obtained with a broad view. With the different colours, it is possible to identify

the expected crop and the type of crop that the model has finally classified, as shown in Figure 12. There are four images. The first is the desired mask, also known as ground truth. The second one is predicted only on the pixels for which it is a ground truth to compare. The third one consists of using the model over all the pixels, even those for which it is not known their ground truth. Finally, there is an RGB image on a summer day to get an overview of the terrain analysed in Figures 13–15. As can be seen, considerable accuracy based on ground truth is clearly shown (Figure 13a), and the plots adjoining the declared crops are detected, giving a broader meaning to the results.

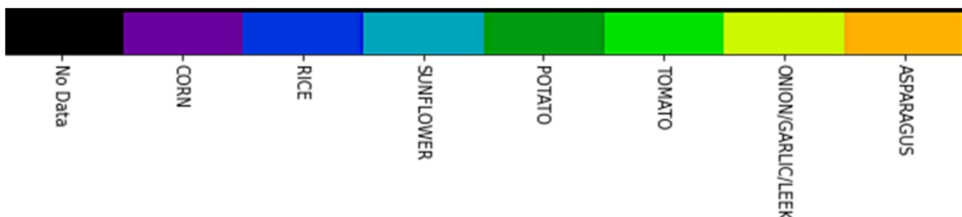

**Figure 12.** Colour scale of categories.

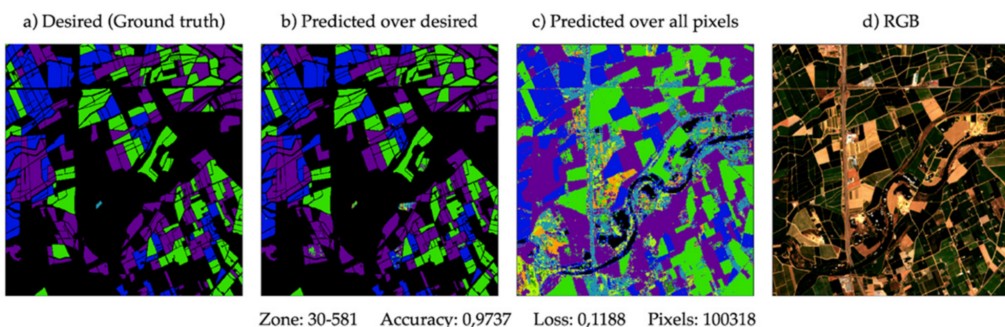

**Figure 13.** Prediction example of zone 30-581.

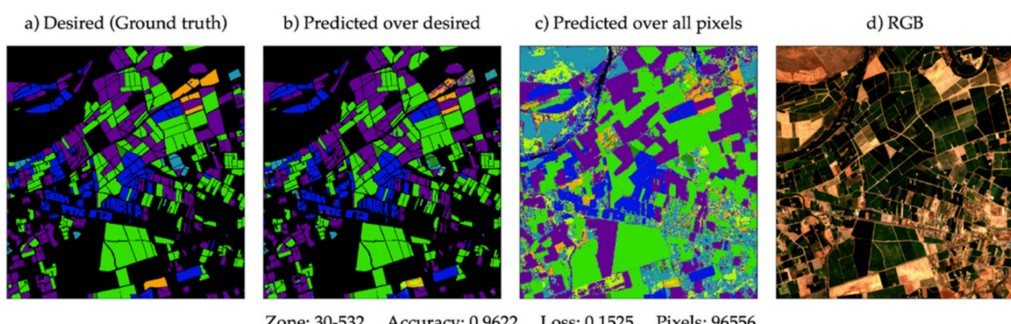

**Figure 14.** Prediction example of zone 30-532.

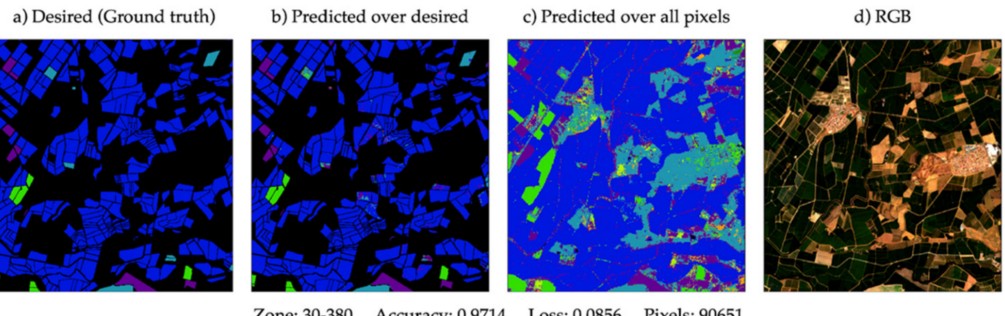

**Figure 15.** Prediction example of zone 30-380.

For final visual validation, all the areas studied are put together in a single image to have an overview of the masks obtained by the model. The images available are: RGB, expected (ground truth), predicted over the expected, predicted over all pixels. Figures 16–19.

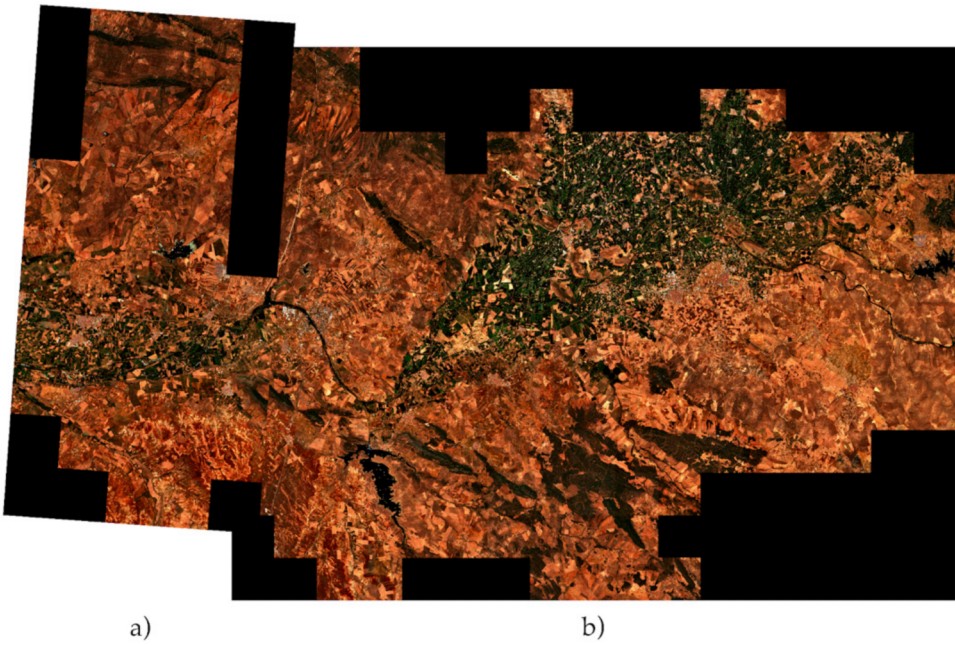

**Figure 16.** RGB of Zone 29 (**a**) and 30 (**b**). Mérida and Don Benito, Extremadura.

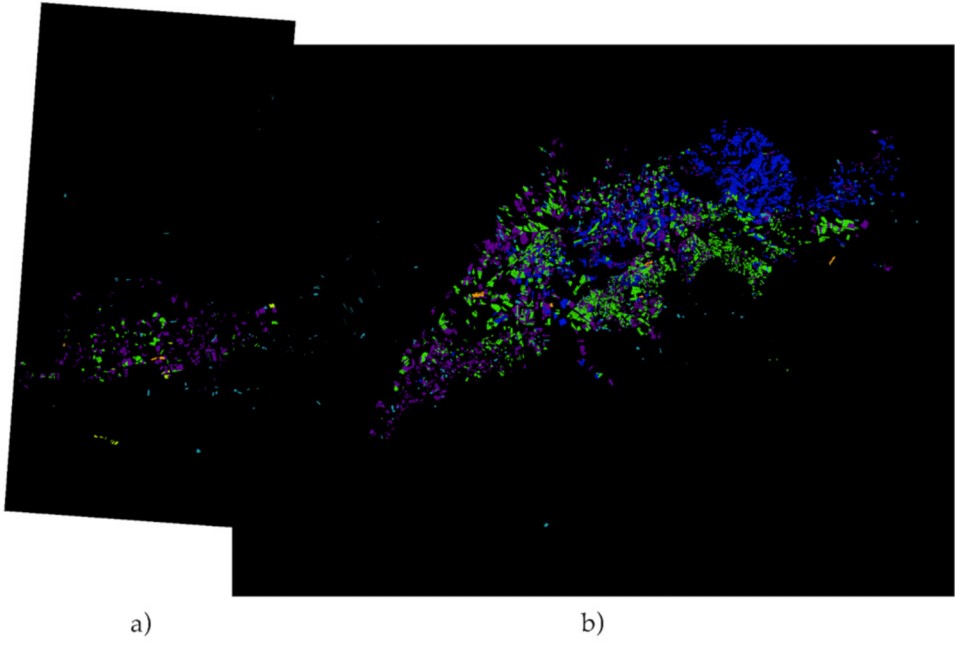

**Figure 17.** Expected (Ground truth) mask of Zone 29 (**a**) and 30 (**b**). Merida and Don Benito, Extremadura.

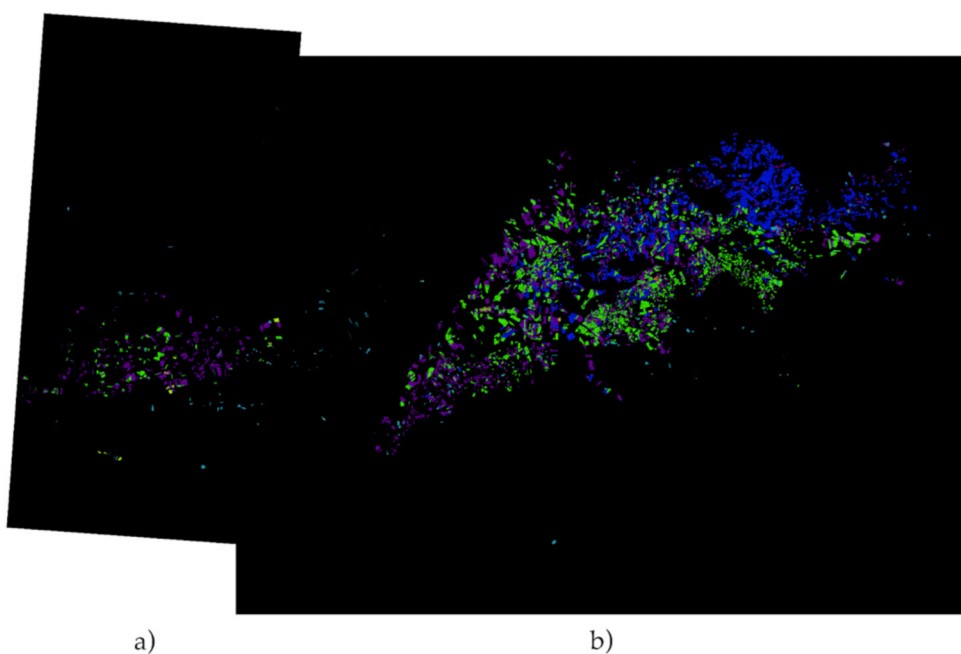

**Figure 18.** Predicted over the expected mask of Zone 29 (**a**) and 30 (**b**). Mérida and Don Benito, Extremadura.

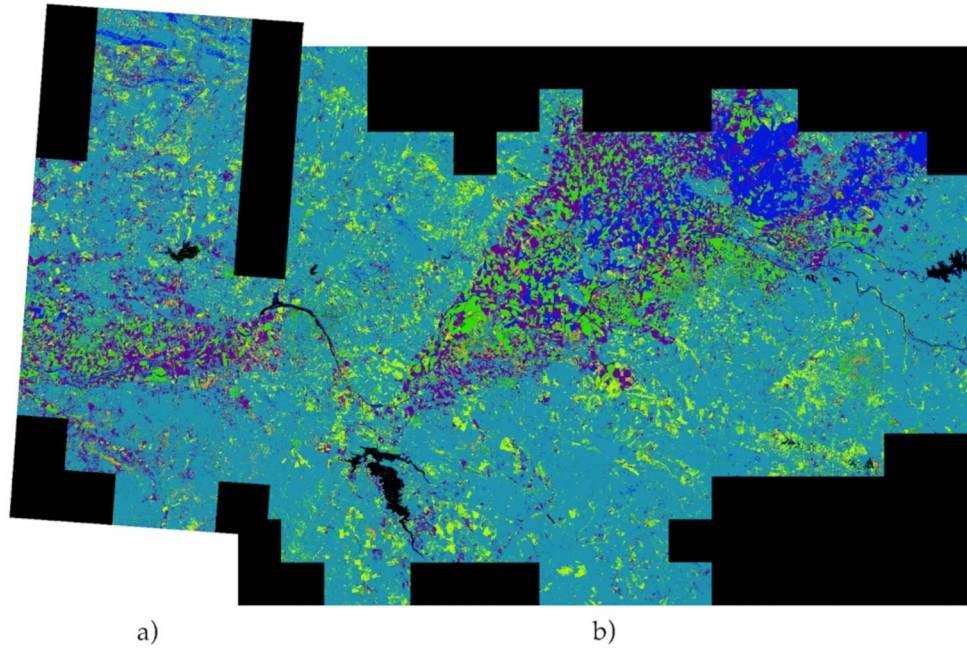

**Figure 19.** Predicted over all pixels mask of Zone 29 (**a**) and 30 (**b**). Mérida and Don Benito, Extremadura.

## 4. Conclusions

This work presents a novel approach for identifying crops that applies neural networks with multiple satellite images using synthetic inputs at the pixel level. The pixel level band information is used for each day that is available in Sentinel-2 images. With this information, the synthetic matrix combines corresponding bands and days as input for the convolutional neural networks. This information is also processed with outliers and interpolation algorithms to account for possible noise, which is primarily due to the atmospheric situation. This methodology was applied to a real-world environment to assist agricultural controls of the Regional Government of Extremadura CAP for the year 2020, classifying the various types of crops sown in summer in an area of Extremadura (Spain).

To date, the Regional Government of Extremadura has carried out arbitrary inspections of the plots claiming that misclassified plots account for only less than 1.43 percent of all plots. For this reason, the few misclassifications are negligible enough to affect the net output result after training.

Using our system, after comparing the predictions with the classified data and arbitrary inspections conducted by the Regional Government, it is clear that there are substantial results (F-score of 96 percent on average), particularly in the dominant crops. Running the prediction through all of the pixels of the zones, including the unclassified areas, reveals that there are adjoining plots that are not present in the datasets but make a lot of sense given the area and location. Additionally, it is a very scalable system that can be tested with various crops, even at different times of the year and in different locations. It is likely that rainier study areas, such as northern Spain, will be more difficult, and other possible solutions could be explored, such as adding information to the synthetic matrix from radar bands from other types of satellites, for example, Sentinel-1 images.

It is noteworthy that the novel approach of the synthetic image system achieves an advantage in training time. True, this necessitates prior data processing, but in this type of problem, it is always necessary due to the requirement of the temporality variable, as a result, a large number of sentinel images must be downloaded and processed. The storage capacity for the data inputs will be the most important determinant of scalability.

With our approach, it is possible to have a more exhaustive control of the situation on the land, improving the statistics of land use, and controlling areas that until now could not be checked due to their difficult access or the impossibility of controlling them with the methods used previously. It is possible to compare production across years, contrast it with climatic factors, and avoid potential risks in the event of drought or abnormal conditions. The results obtained by combining neural networks and synthetic images indicate promising possibilities for carrying out various projects such as those mentioned above.

**Author Contributions:** G.S. and M.F.-S. conceived and designed the framework of the study. G.S. completed the data collection and processing. G.S., M.F.-S. and A.L.-T. completed the algorithm design and the data analysis and were the lead authors of the manuscript. All authors have read and agreed to the published version of the manuscript.

**Funding:** This work was funded by the FEDER inter-administrative collaboration agreement 330/18 between the Junta de Extremadura, Consejería de Medio Ambiente y Rural, Políticas Agrarias y Territorio and Universidad de Extremadura; and by the Junta de Extremadura, Consejería de Economía e Infraestructuras under grant IB18053 and by the European Regional Development Fund (ERDF).

**Institutional Review Board Statement:** Not applicable.

**Informed Consent Statement:** Not applicable.

**Data Availability Statement:** The raw/processed data required to reproduce these findings cannot be shared at this time as the data also forms part of an ongoing study.

**Acknowledgments:** Thanks to Junta de Extremadura, Consejería de Economía e Infraestructuras, Spain for the provided data used in this work.

**Conflicts of Interest:** The authors declare no conflict of interest.

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
