# Peer review of "Crop Classification of Satellite Imagery Using Synthetic Multitemporal and Multispectral Images in Convolutional Neural Networks"

_remotesensing, doi:10.3390/rs13173378_

Round 1

Reviewer 1 Report

Nice paper. I would suggest the author to improve the english language.

Author Response

Thank you very much for your review. The paper has been reviewed by a native English, and she has corrected some expressions.

Reviewer 2 Report

This manuscript aims to explore a better approach to classify crops and distinguish between several crop. It has certain significance. This has played a very important role in the statistics of planting area of different crops in the future. There are still some suggestions for the author. 

The introduction also needs significant improvement. There is not a clear story line towards the objective of this study. For instance, the manuscript uses satellite image for crop classification, and there is no research progress in this field in the introduction. All I can see is the advantages and disadvantages of some classification methods. The author should add relevant contents.

Author Response

Thank you for your review. The introduction has been revised mentioning the objective of the work, and some specific references related to crop classification have been added.

Reviewer 3 Report

The paper “Crop Classification of Satellite Imagery using Synthetic Multi-temporal and Multispectral Images in Convolutional Neural 3 Networks” focuses on the development of a new method that creates synthetic images by extracting  Sentinel 2 satellite data at the pixel level, processing all available bands, as well as their data distributed over time considering images from multiple dates. In this study the authors have applied neural networks with multiple satellite images using synthetic inputs at pixel level. The information from each band and each day embedded as synthetic matrix has been used as input for the convolutional neural networks. This information has been processed with outliers and interpolation algorithms to account for possible noise caused by the atmospheric situation. This methodology was applied to a real-world environment to assist agricultural controls of the Regional Government of Extremadura CAP for the year 2020, classifying the various types of crops sown in summer in an area of Extremadura (Spain). A high accuracy predictions for 12 zones have been obtained by using this new method.

Recommendations for authors: to insert geographical coordinates in Fig. 6, to explain in detail the methodology used, what software was used for CNN predictions and how those 12 zones were selected. It would be nice if you highlight these zones in Fig 6 or other figure.

Author Response

Thank you very much for your review. We have modified the paper with your recommendations. Regarding the 12 zones used in the prediction, it was not clear enough in the paper, and the paragraph has been rewritten. We only represented 12 zones (of 204 zones) in particular in Table 3 sorted by number of pixels in descending order.